# Contrast Enhanced EUS for Predicting Solid Pancreatic Neuroendocrine Tumor Grade and Aggressiveness

**DOI:** 10.3390/diagnostics13020239

**Published:** 2023-01-09

**Authors:** Gianluca Franchellucci, Marta Andreozzi, Silvia Carrara, Luca De Luca, Francesco Auriemma, Danilo Paduano, Federica Calabrese, Antonio Facciorusso, Valeria Poletti, Alessandro Zerbi, Andrea Gerardo Lania, Alexia Francesca Bertuzzi, Paola Spaggiari, Vittorio Pedicini, Marcello Rodari, Pietro Fusaroli, Andrea Lisotti, Andrew Ofosu, Alessandro Repici, Benedetto Mangiavillano

**Affiliations:** 1Gastrointestinal Endoscopy Unit, Humanitas Mater Domini, 21053 Castellanza, Italy; 2Department of Biomedical Sciences, Humanitas University, 20072 Pieve Emanuele, Italy; 3Endoscopic Unit, Department of Gastroenterology, IRCCS Humanitas Research Hospital, 20089 Rozzano, Italy; 4Department of Clinical Medicine and Surgery, ‘Federico II’ University of Naples, 80131 Naples, Italy; 5Endoscopic Unit, ASST Santi Paolo e Carlo, 20142 Milan, Italy; 6Gastroenterology Unit, Department of Medical and Surgical Sciences, University of Foggia, 71122 Foggia, Italy; 7Pancreatic Surgery Unit, Humanitas Clinical and Research Center—IRCCS, 20089 Rozzano, Italy; 8Endocrinology, Diabetology and Medical Andrology Unit, Humanitas Clinical and Research Center—IRCCS, 20089 Rozzano, Italy; 9Medical Oncology and Hematology Unit, Humanitas Clinical and Research Center—IRCCS, 20089 Rozzano, Italy; 10Department of Pathology, Humanitas Clinical and Research Center—IRCCS, 20089 Rozzano, Italy; 11Department of Interventional Radiology, Humanitas University, Humanitas Research Hospital IRCCS, 20089 Rozzano, Italy; 12Department of Nuclear Medicine, IRCCS Humanitas Research Hospital, via Manzoni 56, 20089 Rozzano, Italy; 13Gastroenterology Unit, Hospital of Imola, University of Bologna, 40126 Imola, Italy; 14Division of Gastroenterology and Hepatology, University of Cincinnati, Cincinnati, OH 45221, USA

**Keywords:** EUS, CE-EUS, CH-EUS, PNET, aggressiveness, grading, NET

## Abstract

Pancreatic neuroendocrine tumor (PNET) behavior assessment is a daily challenge for physicians. Modern PNET management varies from a watch-and-wait strategy to surgery depending on tumor aggressiveness. Therefore, the aggressiveness definition plays a pivotal role in the PNET work-up. The aggressiveness of PNETs is mainly based on the dimensions and histological grading, with sometimes a lack of specificity and sensibility. In the last twenty years, EUS has become a cornerstone in the diagnostic phase of PNET management for its high diagnostic yield and the possibility of obtaining a histological specimen. The number of EUS applications in the PNET work-up has been rapidly increasing with new and powerful possibilities. The application of contrast has led to an important step in PNET detection; in recent years, it has been gaining interesting applications in aggressiveness assessment. In this review, we underline the latest experiences and opportunities in the behavior assessment of PNETs using contact-enhanced EUS and contested enhanced harmonic EUS with a particular focus on the future application and possibility that these techniques could provide.

## 1. Introduction

Neuroendocrine tumors (NETs) are a large group of different neoplasia originating from the diffuse neuroendocrine cell system, which can occur in many sites around the body. The gastrointestinal system is the most common site where NETs arise, followed by the lung [1], with pancreatic NET (PNET) representing about 12% of gastrointestinal neuroendocrine tumors (GNETs) [2]. In the last four decades, the global incidence of GNETs has been rising; it has been estimated that the incidence rate increased 6.4-fold from 1975 to 2015 [3], with the majority of cases being diagnosed incidentally at early stages [4]. This trend has raised the need for a better definition of PNET prognosis to define which tumors should be resected or followed in surveillance programs and which should not. All NETs are classified according to their origin site, local and extra local extension (staging), and grading (G1–G2–G3), which is based on the proliferation index (Ki-67%) [5]. NETs can also be defined as functional or non-functional according to their ability to produce active hormones: the first is associated with specific clinical syndrome such as the Zollinger–Ellison one; finally, PNETs can be sporadic or be part of a genetic syndrome, multiple endocrine syndrome (MEN1) is the most common entity [6].

The current management of PNETs is highly dependent on the disease stage and grade [1], which may result in a watchful waiting strategy or surgical resection [1,7,8]. Therefore, when clinicians encounter a PNET, the primary question they must answer is “how aggressive is the tumor?”. The current diagnostic work-up with computed tomography (CT), magnetic resonance (MRI), and endoscopic ultrasound (EUS) can detect a NET smaller than 1 cm with good accuracy. Once the tumor is detected, its aggressiveness is primarily determined by its size [9,10] and grade [11]: while the size can be easily quantified with modern radiologic tools, the current standard for grading assessment is analyzing the material obtained from EUS-guided fine needle aspiration (EUS-FNA) [12,13,14,15]. Ki-67 and tumor differentiation have been seen to have a greater impact on survival than the disease stage [16], so their definition plays a key role in PTEN assessment. The evidence of a high grade of fibrosis and microvascular density decrease (MVD) on tumor pathological specimens has led many authors to search for undirected malignant features of PNET extrapolated from non-invasive techniques (CT, MRI) as a macroscopic translation of a microscopic language [17,18,19]. Going in this direction, EUS application with its high sensibility and specificity could be a reliable tool to assess PNET behavior other than typical morphological features.

Recently, the use of elastography, contrast-enhanced EUS (CE-EUS), and contrast-enhanced harmonic (CH-EUS) has highlighted new characteristics with prognostic value that may aid in complex clinical decision-making in the NET field. In this mini-review, we will discuss current EUS applications in the diagnosis of solid PNET diagnosis, with a focus on CE-EUS and CH-EUS for defining tumor grading and aggressiveness.

## 2. EUS in Solid Pancreatic Neuroendocrine Tumor

Nowadays, EUS represents a cornerstone in the diagnosis and localization of gastrointestinal NET [20]. In most cases, it is the only technique that can provide a definitive preoperative diagnosis. Actually, EUS has demonstrated superior accuracy compared to CT scan for detecting very small PNETs, particularly insulinomas [21]. In a recent metanalysis on 612 patients, it was estimated that preoperative EUS assessment could increase the detection rate of about 25% [2].

On EUS, basal analysis PTENs typically appear as homogeneous hypoechoic lesions with regular margins (Figure 1); sometimes they can have a cystic appearance [22] while in the most advanced cases, they can lose these characteristics, acquiring features more similar to pancreatic adenocarcinoma [23].

In addition to detecting NETs, the emerging role of EUS is to provide information about tumor locoregional staging, grading, and prognosis. While tumor size alone is not a reliable indicator of non-aggressiveness, it is also be seen as a recurrence risk factor post-surgical resection [24], and certain EUS findings can be significant predictors of malignancy.

Indeed, even though large PNETs are suggestive of malignancy because they are likely to be associated with local or vascular invasion, some cases of PNET <2 cm with angioinvasion or lymph node metastasis have been described [25]. As malignant PNETs are inclined to hemorrhage, hyalinosis, necrosis, and cystic change, a heterogeneous ultrasonographic texture and hypoechoic and anechoic areas must be considered carefully as they probably correspond to hemorrhage and necrosis areas on pathological examination. In a retrospective analysis [26] of 41 PNETs, the heterogeneous appearance at EUS was identified as the characteristic most strongly associated with malignant behavior, even more than tumor size. Obstruction of the main pancreatic duct can also be seen on EUS; however, it is not necessarily indicative of malignancy because a benign tumor developing near the duct may cause obstruction by expansive pressure, despite its size [26]. The obstruction of the main duct was also not correlated with a malignant behavior in the study of Pais [27], where irregular margins and greater diameter were significantly related to malignant evolution. In the retrospective studies of Fujimori [14], main pancreatic duct obstruction (Odd ratio:2.8 *p* = 0.02) and heterogeneous texture (*p* = 0.01) on EUS were defined as significative predictors of G2/G3 histology. The results of these different experiences may appear controversial; in reality, it is hard to compare studied with a huge difference in the definition of malignancy and aggressiveness of PTEN; while Fujimori [14] considered aggressive G2/G3 tumor as the definition on 2010 PTENs by the WHO, Ishikawa [26] and Pais [27] considered aggressiveness according to the 2004 WHO PTEN classification.

To date, EUS-FNA represents the gold standard for assessing the proliferation index (Ki-67%), which defines the tumor grade and is strongly associated with prognosis and a therapy approach [28,29]. EUS-FNA showed good concordance (between 58% and 78%) with surgical specimens [28,29,30], although a non-negligible rate of under-rating when in the tissue count with more than 2000 cells [31,32]. In a recent study, Crinò et al. [31] retrospectively compared the fine-needle-biopsy (EUS-FNB) and the EUS-FNA in the Ki-67 assessment and they found a stronger Ki-67 concordance between EUS-FNB and the surgical specimens than between EUS-FNA and the surgical specimens (96.1% vs. 88.2%, respectively; *p*: 0.04).

In a recent meta-analysis on 864 patients who underwent EUS-FNA/FNB and surgical resection for PNET, the overall concordance between EUS grading (eG) and surgical grading (sG) was estimated to be 80.3%, under-grading occurred significantly more frequently than over-grading. These data confirmed that EUS is an accurate technique in defining PNET grading, despite the presence of a margin of error and the possibility of over- or undertreatment [33]. In addition, the grade does not always reflect tumor aggressiveness, and sometimes G1 or G2 tumors can present with distant metastasis [25,34]. Therefore, more efforts are required to identify additional characteristics that can predict the aggressiveness of PNET and guide therapeutic strategies. Recent studies [35,36] have started to perform EUS with the application of deep-learning in the assessment of pancreatic masses. AI could improve the diagnostic yield and be a very useful tool to reduce the operator-dependent characteristics of EUS, and no studies have yet been performed on the characterization of PTEN.

## 3. Role of Elastography in Pancreatic Neuroendocrine Tumor

From the first study conducted in 2006 by Giovannini on 49 patients [37], elastography has been widely used in the last years for increasing the detection performances of EUS in assessing solid pancreatic lesions. Elastography defines the parenchyma stiffness in two different types of analysis: one qualitative and one semiquantitative. The qualitative analysis defines the different stiffness of the analyzed tissues according to a colorimetric scale. In the semiquantitative analysis, a numeric value of the strain ratio between two regions of interest (i.e., the lesion and the surrounding parenchyma) is calculated, and a strain histogram of a certain and well-defined region of interest [38] can be reproduced.

The different metanalysis conducted in the last ten years [39,40] reported a sensibility over 90% for both qualitative and semiquantitative elastography and a sensibility between 60–70% in the differentiation of benign vs. malign lesions. Given the low frequency of PNET, the largest amount of data on elastography referred to pancreatic adenocarcinoma, but some interesting data could be indirectly picked out and applied to PNET.

Elastographic characterization has provided different, heterogenous, and sometimes controversial evidence. In 2009, Iglesias-Garcia et al. [41] conducted the first significative prospective analysis on the qualitative elastography pattern of solid pancreatic lesions. They divided the elastography pattern of the lesion of interest in four color regions: from the harder to the softer tissues, they assigned the following colors, respectively: dark blue, green, yellow, and red. All the PNETs diagnosed had a blue pattern at the elastography analysis and all of them were defined as malignant at the pathological analysis. The same group of authors [42] also performed a prospective study with the aim of analyzing the semiquantitative characteristics of PTEN on elastography and they observed that PNET had the highest strain ratio (52.1, CI, 33.96–70.71) among the pancreatic solid lesions including pancreatic adenocarcinoma, chronic pancreatitis, and other inflammatory pancreatic conditions (Figure 2).

However, these results were questioned by other subsequent experiences: Ignne et al. [43] conducted a multicenter prospective study, with a sample size of 114 pathologically diagnosed PNETs, demonstrating that 64% of them were softer than the surrounding parenchyma. A similar study [44] observed that PNET had a significative lower strain ratio than malignant solid lesions. Concordant results came from a study conducted by our group [45] in 2018, where PNETs showed a significative (*p* < 0.001) lower average lesion-to-parenchyma strain ratio (7.1, CI, 3.5–11.2;) and lower lesion-to-wall strain ratio (14.1, 95% CI, 6.24–21.9) compared to malignant solid lesions, although no significative differences were observed between PNETs and other benign solid lesions. The study was conducted with an eco-processor EU-ME (Olympus, Europa SE & Co KG, Hamburg, Germany), unlike the other mentioned studies where different kinds of Hitachi (Medical Systems Europe, Zug, Switzerland) processors were used to perform EUS. Furthermore, these data were [45] relevant for the computer aided-fractal analysis. Fractal geometry is in summary a mathematical tool for describing the roughness of objects; the application of this tool for PNET analysis shows that there were significative differences in the mean surface fractal dimensions between NET and malignant lesions such as pancreatic adenocarcinoma, surrounding regular parenchyma, and inflammatory lesions (*p* < 0.087).

From all the mentioned evidence, it is still not possible to define a univocal cut-off for both the differentiation between benign and malignant lesions in general and for the definition of the PNET nature. Actually, searching for a PTEN elastography cut-off was not the main aim of these studies, which all suffered from small sample sizes and heterogeneous features, first, the technical differences between the instruments used for elastography analysis were investigated.

## 4. Contrast-Enhanced EUS and Contrast Harmonic EUS for Pancreatic Neuroendocrine Tumor: The Assessment of Aggressiveness

Together with technological advancements, the introduction of contrast-enhanced EUS (CE-EUS) has dramatically improved the resolution and the application of EUS in the NET field. CE-EUS was first studied by Kato in 1995; it is based on the administration of microbubble-based contrast-enhanced agents (air filled-bubbles of 2–5 µm with a lipids of phospholipids shell) [46] during EUS examination, which highlight different enhancement patterns in pancreatic lesions reflecting their vascularity.

The PNETs typically show hyperenhancement (Figure 3) on CE-EUS, even when very small in size, and are therefore difficult to detect on basal scans [47].

A prospective analysis in 2008 on 93 patients [48] with solid pancreatic lesions smaller than 40 mm showed that a hypovascular pattern at CE-EUS had a sensibility of 92% and specificity near 100% in differentiating pancreatic adenocarcinoma vs. other benign lesions. In their sample, there were 50 PNET; of them, only 20 fulfilled the inclusion criteria for the study and they all had a hypervascular appearance at CE-EUS.

In 2010, there was the first attempt to properly define CE-EUS utility in differentiating malignant and benign PNET. Ishikawa and colleagues [26] correlated the CE-EUS characteristics of 41 patients with a PNET diagnosis confirmed by surgery and or by EUS-FNA with benign or malignant tumor behavior. They arbitrary categorized the contrast enhancement in three different patterns: diffuse enhancement (A), filling defects (B), no enhancement (C). On 41 PNETs, 40 (97.6%) showed obvious enhancement, and 18/21 malignant PNETs (85.7%) showed a type B CE-EUS pattern. Judging type B and C patterns as malignant, the sensitivity, specificity, and accuracy of CE-EUS for malignancy were 90.5%, 90.0%, and 90.2%, respectively. Additionally, CE-EUS was able to highlight necrotic and hemorrhagic lesions, typical of advanced tumor, which were identified as a contrast-filling defect with better sensibility compared to conventional EUS.

MVD is a histological report, classically measured as the number of CD31-positive vessels/mm^2^ and its decrease was already related to a negative behavior of PNET [49,50,51]. A recent retrospective [52] monocentric analysis evaluated the micro vessel density (MVD) on the PNET surgical specimen and its correlation with tumor aggressiveness. Specifically, this study underlined a correlation between a decreased MVD and a higher frequency of nodal metastasis (0.16). As the second end point, the study evaluated the ability of CE-EUS to assess the MVD. Thirty-six patients were enrolled and arterial hypo-enhancement (*p*: 042) and late enhancement washout in CE-EUS (*p*: 0.34) were seen to be significatively related to decreased MVD in the surgical specimens.

Contrast-enhanced harmonic EUS (CH-EUS) is a novel technique that allows for the visualization of the micro-vascularization in the pancreas and pancreatic tumors [28] while avoiding color and power Doppler artifacts.

Contrast harmonic imaging detects signals from microbubbles and filters signals that originate from the tissue by selectively detecting the harmonic components. This technology can detect signals from microbubbles in vessels with a very slow flow so the technique can more precisely assess the microcirculation of the interested parenchyma by selectively detecting the harmonic components of signals from contrast microbubbles in the vessels [53].

In 2010, the first pilot study on the application of CH-EUS on pancreatic solid lesions was conducted by Napoleon et al. [54] on 35 patients; they assessed high sensitivity, specificity, positive predictive value, negative predictive value, and accuracy (89%, 88%, 88%, 89%, and 88.5%, respectively) for the detection of pancreatic adenocarcinoma.

In a similar study by Kitano et al. [55] on NETs, CH-EUS reached a sensitivity and specificity of 78.9% and 98.7%, respectively, in the diagnosis of hypervascular NET.

Later in 2018, Palazzo et al. [56] evaluated the efficacy of CH-EUS in predicting PNET aggressiveness. Using a linear echoendoscope and a second-generation US contrast agent (Sonove; Bracco, Milan, Italy), they collected the characteristics of various lesions between 25 and 45 min after contrast injection. The aggressiveness of CH-EUS tumors was characterized by heterogeneous enhancement during the early arterial phase. The definitive tumor aggressiveness was determined by the World Health Organization (WHO) classification [57] as G3 tumors or morphologic and/or histologic findings of metastatic disease in G1/G2 tumors. In 35 of the 81 collected cases, CH-EUS revealed heterogeneous enhancement and were classified as aggressive based on histological analysis.

The accuracy, sensitivity, specificity, positive predictive value (PPV), and negative predictive value (NPV) of CH-EUS in determining tumor aggressiveness were 86%, 96%, 82%, 71%, and 98%, respectively.

Heterogeneous enhancement at CH-EUS corresponded to pathologic specimens with fewer vascular and more fibrotic tumors.

Notably, CH-EUS demonstrated superior accuracy compared to “classical parameters” such as the Ki-67% and the tumor size (>2 cm) in defining PNET aggressiveness, particularly in G1/G2 tumors without visible metastasis. In this study, the good interobserver agreement between the endoscopists who evaluated the enhancement characteristics while blinded to patient history radiologic, histologic, and surgical reports was also noteworthy.

Following Palazzo [56], the Ishiwaka group [58] conducted a retrospective study in 2021 correlating the CH-EUS appearance of solid PNET with their clinical aggressiveness. They considered three different contrast patterns: iso-, hypo-, and hyper-enhancement. Of 47 tumors, 19 were characterized as aggressive; hypo-enhancement at CH-EUS was identified as an indicator of aggressiveness, demonstrating a sensibility, specificity, PPV, NPV, and accuracy of 94.7%, 100%, 100%, 96.6%, and 97.9%, respectively. Hypo-enhancement at CH-EUS was also associated with a worse prognosis in G1/G2 tumors, with only one aggressive tumor on histology that did not show hypo-enhancement on CH-EUS. Additionally, in this case, hypo-enhancement on CH-EUS was associated with a greater degree of fibrosis and smaller and fewer vessels in the resected specimens with a reduction in MVD strongly associated with tumor aggressiveness (*p* < 0.001).

These results are in line with the study of Battistella [52] in which the authors added a negative prognostic value to the late arterial washout. Reduced MVD and exceeding fibrosis are well-known pathological negative prognostic factors of PNET behavior; all findings in CE/CH-EUS significantly correlated with increased aggressiveness or advanced stage at the diagnosis [49,50,59], which represent in synthesis a macroscopic translation of these histological features. The exposed data on the correlation between PTEN endoscopic appearance in CE/CH-EUS and tumor characteristics were similar to previous experiences conducted by the radiologist in the first decade of the 21st century. All of the correlations they found on the appearance of PTEN during CT contrast assessment were related to both a reduced MVD of the tumor and an increased presence of fibrosis [17,18,19].

All of the above-mentioned studies are based on a qualitative assessment of the EUS appearance of a solid PNET, which embrace the challenge of the dependence on the EUS operator. 

CH-EUS provides a real-time image of the micro-vascularization of the pancreatic parenchyma using the temporal change in echo enhancement intensity, which can be measured and expressed by the time intensity curve (TIC), a quantitative parameter that was previously used in breast, renal, and liver tumors [60,61].

In 2017, Omoto and colleagues [62] found that pancreatic adenocarcinoma and PNET showed significative differences (*p* < 0.005) in two TIC parameters: the values of peak intensity and the intensity at 60 s after contrast injection; these two parameters were significative lower in pancreatic adenocarcinoma than in PTEN (*p* < 0.05).

Takada and colleagues [63] went further and studied the TIC parameters to find the potential predictor data about PNET aggressiveness.

In their study, two experienced endoscopists performed all the procedures on a sample of 30 patients. The analysis was recorded within a 2 min window following contrast administration (Sonazoid, Daichii-Sankyo, Tokyo, Japan). The two regions of interest were the solid lesion and the normal surrounding pancreatic parenchyma. The TIC parameters examined were: (I) echo intensity change; (II) time for peak enhancement, (III) speed of contrast; (IV) decrease rate for enhancement; (V) enhancement ratio for node/pancreatic parenchyma. Among them, the echo-intensity change, time for peak enhancement, and enhancement ratio for node/pancreatic parenchyma demonstrated a significant relationship with tumor grading (Table 1).

A recent study [64] tried to correlate some TIC parameters to the overall survival in PNET and pancreatic adenocarcinoma. The authors found a significative correlation between lower peak enhancement (HR = 1.76, *p* = 0.02) and lower wash area under the curve (HR = 1.06, *p* = 0.001) with pancreatic adenocarcinoma. No significative relationship emerged with PNET due to the very limited size of the sample, which was only eight cases.

The study results added important information to the definition of PNET aggressiveness.

Ishiwaka [58] and colleagues first recognized the higher sensitivity of EUS in the detection of PNET (95%) compared to CT scan (80.6%) and abdominal US (45.2%). They also demonstrated the heterogeneous contrast enhancement and the absence of contrast enhancement on CE-EUS as malignant features in PNET.

These results have been corroborated by other studies by Palazzo [56] and Ishiwaka [58] on CH-EUS. In addition to CE-EUS, the presence of a heterogeneous contrast enhancement was significantly associated with malignant behavior on CH-EUS.

This step was taken to eliminate a potential confounding variable in the study of Palazzo, in which the heterogeneous enhancement group included characteristics with opposite prognostic significance: the cystic degeneration [65,66,67] and calcification [68] of the parenchyma.

Since EUS is highly operator dependent, it is important to notice the excellent operator concordance reported by Palazzo et al. [56], where the intra-observer and inter-observer agreement, expressed in K coefficient, was >0.80 for both the senior and junior endoscopists. No references on concordance were made by Ishikawa et al. [26,58] in both of their studies.

It is also important to note that all of the procedures analyzed were performed by very skilled endoscopists from both the Palazzo [56] and Ishikawa groups [58], who achieved sensitivity and specificity close to 100% in the PNET aggressiveness assessment. This characteristic, along with the small sample size, may have led to an overestimation of the excellent results obtained and may have an impact on the reproducibility of the studies and the clinical application of CH-EUS.

Takada et al. [63] studied the relationship between the time–intensity curve (TIC) obtained from CH-EUS and the histological tumor grade to develop a quantitative model to define the aggressiveness of PNET for the first time.

With this method, they achieved a good level of sensitivity and specificity (>90%) in differentiating between G1/G2 and G3 tumors, albeit with less sensitivity in predicting the aggressiveness of the G1 and G2 neoplasia.

The first limitation of their study was that not all the PNETs had a histologic report on the surgical specimens, whereas all the G3/neuroendocrine carcinoma had only EUS-FNA as the definitive diagnosis, with the inherent limitations associated with the method [26].

The second limitation is that they could only include cases in which TIC measurement was feasible, thus reducing the size of the study population and the power of the study.

In the end, all of the studies mentioned were conducted retrospectively, with single or dual-center experiences. Only one study had a follow-up period, and all studies had a small sample size, the largest one including 81 patients.

Finally, it is worth mentioning that some authors have tried to study the detection yield of a combination of all the techniques already mentioned, CE-EUS, CH-EUS, and elastography. In 2017, Inglesias-Garcia [69] performed a prospective study to evaluate the combination of CH-EUS and elastography in the definition of solid pancreatic lesions. They collected 64 patients and compared the accuracy in defining the malignant behavior of CH-EUS, quantitative elastography, a combination of both techniques (CH-EUS and elastography), and EUS FNA with the following results: 98.4%, 85.5%, 91.9%, and 91.5% (95% CI: 83.6–99.5), respectively. In this study, only three PNET cases were encountered, of which only one showed a malignant pattern, with a hypervascularity enhancement in CH-EUS and a high strain ratio (about 52) assessed with a Hitachi processor. However, this evidence have been questioned by other authors who consider the definition of a CH-EUS hyper vascular pattern as a marker of malignant disease controversial, since other benign entities such as autoimmune pancreatitis could show this enhancement pattern [70].

Other studies [71,72] have tried to compare the application of CE-US and semiquantitative elastography in the detection and assessment of the malignancy behaviors of pancreatic masses, but all of the studies established neuroendocrine nature as an exclusion criteria.

## 5. Conclusions and Future Prospective

In conclusion, EUS has a significant role in the approach to PNET, not only being a diagnostic tool, but also a predictive instrument of tumor aggressiveness. EUS and EUS-guided-sampling have already affirmed their relevance in PNET localization, staging, and grading, therefore guiding both therapeutic and follow-up strategies. It is clear that the improvement in the “art” of EUS-tissue acquisition [73,74,75], moving from needle aspiration (FNA) to needle biopsy (FNB), has further promoted its role in the pre-treatment algorithm of PNET.

Furthermore, in this mini review, we highlight new different modalities associated with basal EUS scans such as qualitative and semiquantitative elastography, CE-EUS, and CH-EUS could securely increase the diagnostic yield in PTEN diagnosis, providing relevant information on disease aggressiveness. The most recurring features emerging from the latest evidence in this field are the increased tissue stiffness in elastography and the heterogeneous and hypo vascular patterns in both CE-EUS and CH-EUS as malignant and negative prognostic factors in PNET. These characteristics in the EUS analysis seem to well reflect the heterogeneity, high grade of fibrosis, and decreased micro vascularity (MVD) found on the surgical specimens of malignant PNET.

Certainly, further studies with prospective design are necessary to build reliable data about the EUS prediction of PTEN aggressiveness. Indeed, the available data are often the results of secondary analysis in studies that have primarily focused on pancreatic adenocarcinoma. Consequently, the first purpose should surely be to increase the studied population size. Additionally, one issue is the EUS operator dependence and the technical differences between the echo processors and the specific modalities used, which make it complex to uniform and generalize the current data.

Among the pancreatic diseases, PNETs remain challenging to manage, and given their increasing incidence, we believe they could be the goal of future research and the optimal and future application of newer modalities such as artificial intelligence (AI) in the EUS field.

## Figures and Tables

**Figure 1 diagnostics-13-00239-f001:**
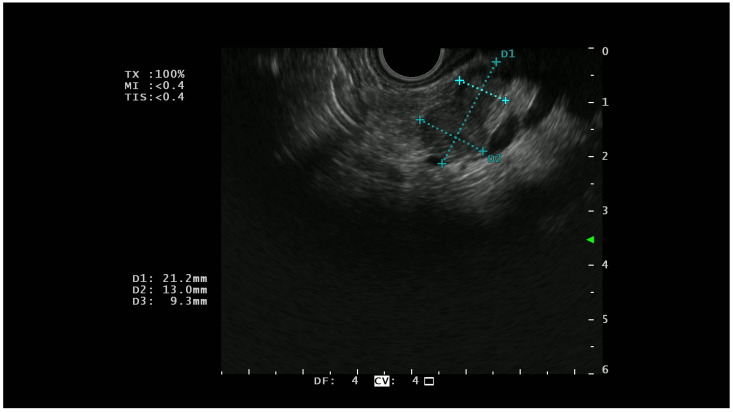
A hypoechoic lesion of 21 mm with regular margins (identified by the blu +) and an ipoechoic appearance; at histopathological analysis the lesion was a G1 NET.

**Figure 2 diagnostics-13-00239-f002:**
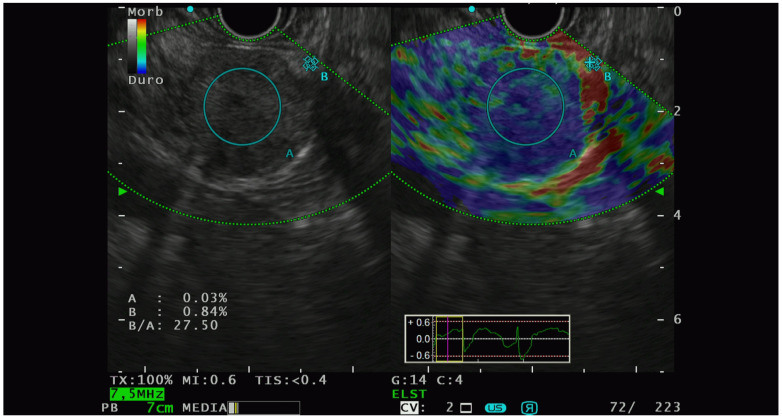
PNET with high lesion-to-parenchyma strain ratio, characterized as G2 with lymphatic invasion at the histological analysis of the surgical specimen.

**Figure 3 diagnostics-13-00239-f003:**
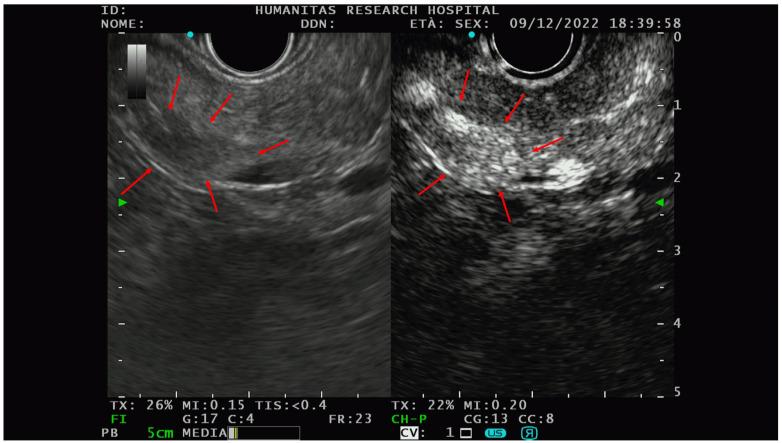
An example of the PNET EUS appearance with and without contrast (Sonovue). In the right picture, there is a hypo-isoechoic lesion while in the picture on the left, the hyperenhancement of the same lesion could be appreciated after intravenous contrast administration. Red arrows delineate the edge of the lesions. After FNAB, the solid lesion was characterized as G1-PNET.

**Table 1 diagnostics-13-00239-t001:** Negative prognostic factors of PTENs in CE/CH-EUS. CE-EUS: contrast-enhanced endoscopic ultrasonography, CH-EUS contrast-enhanced harmonic endoscopic ultrasonography, MVD: micro vascular density, NEC: neuroendocrine carcinoma.

PNET Negative Prognostic Factors	Analysis	Author	Contrast	EUS Assessment	Sensibility (%)	Specificity (%)	Accuracy	Malignant/Aggressive Definition	*p*
Hypoenhancement	Retrospective	Ishiwaka 2010 [26]	Levovist/Sonazoid	CE-EUS	90.5%	90.0%	90.2%	More advanced that uncertain behavior according to WHO 2004 classification	/
Arterial Hypoenhancement	Retrospective	Ishiwaka 2021 [58]	Sonazoid	CH-EUS	94.7	100	97.9	G3/NEC or lymphatic or distant metastasis WHO 2017	<0.01
Heterogenous enhancement	Retrospective	Palazzo 2018 [56]	Sonovue	CH-EUS	96	82	86	G3/NEC or lymphatic or distant metastasis WHO 2017	<0.01
Echo intensity change	Retrospective	Takada 2019 [63]	Sonazoid	CH-EUS	100	96.2	96.7	G3/NEC or lymphatic or distant metastasis WHO 2017	0.0099
Decrease rate for enhancement					100	100	100	G3/NEC or lymphatic or distant metastasis WHO 2017	0.0087
Enhancement ratio for node/pancreatic parenchyma					100	100	100	G3/NEC or lymphatic or distant metastasis WHO 2017	0.2979
Arterial Hypoenhancement	Retrospective	Battistella 2022 [52]	Not specified	CH-EUS	/	/	/	Decreased MVD	0.042
Late arterial washout				CH-EUS	/	/	/	Decreased MVD	0.034

## Data Availability

Not applicable.

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
