# Peer review of "Contrast Enhanced EUS for Predicting Solid Pancreatic Neuroendocrine Tumor Grade and Aggressiveness"

_diagnostics, 2023, doi:10.3390/diagnostics13020239_

Round 1

Reviewer 1 Report

Dear Dr. Gianluca & the Group,

I was asked to provide a review comments on the included manuscript.  Can you provide some examples with color images of the lesions which would enhance the quality of the manuscript.

Best wishes

Author Response

We have appreciated your advice; we added some signature images that could enhance the quality of the abstract.

Reviewer 2 Report

Thank you for your excellent work.  CE-EUS may be a reasonable modality of determining the diagnosis and aggressiveness of P-NET, although histological assessment is definitely required.   I would like you to put some informative points to make more CE-EUS familiar to the readers.

1. Please use some EUS pictures or show some schema when EUS detects PNET aggressive if available. 

2. I understand some contrast media for CE-EUS are available in the market. Please clarify what contrast media the authors in Table 1 used for their research.

Thank you 

Author Response

We have deeply appreciated your advices;

1 We added some images explicative of the contrast-enhanced EUS,

2 We have inserted a new column with the specification of the contrast used in the paper.